# Protein-Protein interactions uncover candidate 'core genes' within omnigenic disease networks

**Abhirami Ratnakumar**[1]*, **Nils Weinhold**[1], **Jessica C. Mar**[2☯], **Nadeem Riaz**[1☯]

**1** Department of Radiation Oncology, Memorial Sloan Kettering Cancer Center, New York, New York, United States of America, **2** Australian Institute for Bioengineering and Nanotechnology, University of Queensland, Brisbane, Australia

☯ These authors contributed equally to this work.
* abhirami.ratnakumar@gmail.com

**Data Availability Statement:** All data that was used in this manuscript are already publicly available from the GWAS catalog (https://www.ebi.ac.uk/gwas/) and the STRING network (https://string-db.org/). All results generated from the

## Abstract

Genome wide association studies (GWAS) of human diseases have generally identified many loci associated with risk with relatively small effect sizes. The omnigenic model attempts to explain this observation by suggesting that diseases can be thought of as networks, where genes with direct involvement in disease-relevant biological pathways are named 'core genes', while peripheral genes influence disease risk via their interactions or regulatory effects on core genes. Here, we demonstrate a method for identifying candidate core genes solely from genes in or near disease-associated SNPs (GWAS hits) in conjunction with protein-protein interaction network data. Applied to 1,381 GWAS studies from 5 ancestries, we identify a total of 1,865 candidate core genes in 343 GWAS studies. Our analysis identifies several well-known disease-related genes that are not identified by GWAS, including *BRCA1* in Breast Cancer, Amyloid Precursor Protein (*APP*) in Alzheimer's Disease, *INS* in A1C measurement and Type 2 Diabetes, and *PCSK9* in LDL cholesterol, amongst others. Notably candidate core genes are preferentially enriched for disease relevance over GWAS hits and are enriched for both Clinvar pathogenic variants and known drug targets—consistent with the predictions of the omnigenic model. We subsequently use parent term annotations provided by the GWAS catalog, to merge related GWAS studies and identify candidate core genes in over-arching disease processes such as cancer—where we identify 109 candidate core genes.

## Author summary

A recent theory suggests that only a small number of genes underpin the biology of a disease, these genes are called 'core genes', and for most diseases, these core genes remain unknown. The suggested methods for finding them requires complex and expensive experiments. We reasoned that if we merge currently available datasets in smart ways, we may be able to uncover these 'core genes'. Our method finds "hub" proteins by merging lists of genes previously linked with disease to information on how proteins interact with

analysis are provided in the Supporting Information tables. Scripts used in this paper are available at https://github.com/AbhiRatnakumar/omnigenic_ ppi_core_genes_plos_genetics_paper.

**Funding:** The authors received no specific funding for this work.

**Competing interests:** The authors have declared that no competing interests exist.

each other. We found that many of these hub proteins have central roles in disease, such as insulin for both A1C measurement and Type 2 Diabetes, BRCA1 in Breast cancer, and Amyloid Precursor Protein in Alzheimer's Disease. We think these 'hub' proteins are candidate 'core genes', and offer our method as a way to find 'core genes' by utilizing publicly available reference datasets.

## Introduction

Genome-wide association studies (GWAS) have uncovered important insights into the genetic basis of complex traits. However, contrary to initial expectation, only a few large effect size-variants with mechanistic links to disease have been identified from these studies [1–8]. The omnigenic model [9] attempts to resolve this paradox by viewing diseases as networks. 'Core genes' which have direct effects on pathways central to pathogenesis (e.g. synaptic genes in schizophrenia [9]) are located at the center of the network, while peripheral genes contribute to disease risk via their influence on core genes. The difficulty of connecting genes within GWAS loci (GWAS hits) to mechanisms responsible for pathogenesis of disease have hindered efforts to understand disease etiologies and develop new therapies [9,10].

Network inference and exome sequencing to identify deleterious, rare variants, have been proposed for identifying core genes [9] directly, although these methods have limitations. For example, the sample size required for exome sequencing to identify rare variants remains uncertain [11] and network inference methods require further technical development [9].

An increasingly accepted view is that diseases can be thought of as perturbations to a network of genes [12]. Intriguingly, functionally related proteins have been shown to reside close within the network [12–14] and a number of studies have leveraged known disease associated genes to find new candidate genes [12]. For example, Markov Random Fields have been used to leverage the topology of pathways to re-prioritize GWAS hits [15]. Support vector machines have been used to learn network features of GWAS hits within tissue specific networks and compute a probability of disease association in all genes [16]. Finally, protein-protein interactions (PPI) with known disease genes have been used to find new disease genes [13].

Although a number of prior studies have integrated GWAS data with networks [16–22], the majority of these studies tended to focus on single diseases or phenotypes [23–26]. The potential for improved insights by integrating GWAS data with protein-protein interaction networks in particular, comes from the finding, that the protein products of genes within GWAS loci physically interact with each other more than expected by chance [27].

Here, we build on prior methods to combine GWAS hits with PPI networks, within the conceptual framework of the omnigenic model, to develop a 'guilt-by-protein-interaction' approach to nominate candidate core genes across a broad range of diseases and phenotypes. We hypothesize that core genes are likely to have excess protein-protein interactions with GWAS hits, and propose a method that combines GWAS hits with the STRING Protein-Protein Interaction network [28], to find proteins enriched for PPI with GWAS hits. In depth examination of the proteins enriched for PPI (PPI genes) reveals properties consistent with 'core genes' as defined by the omnigenic model, and provides insight into disease biology.

## Results

We downloaded the STRING network [28] and extracted physically binding protein-protein interactions with score $> = 700$ (mean score 277.65, range 150–999, **S1A Fig**) yielding a network with 11,049 nodes and 163,181 protein-protein interactions (**S1 Table**) (Methods). We

then extracted GWAS hits from the GWAS catalog [29], and after performing QC and filtering for proteins present in the STRING network (11,049 nodes), identified 3,615 GWAS hits in 1,381 GWAS studies from 5 ancestries (African American or Afro-Caribbean, East Asian, European, Hispanic or Latin American and South Asian (**Fig 1A**, **S2 Table**)).

## Identifying GWAS studies with excess PPI between GWAS hits

We first sought to extend prior work demonstrating GWAS hits have more protein-protein interactions with each other than expected by chance [27]. For each GWAS study, we determined if the number of PPI edges between its GWAS hits on the STRING network exceeded the number of PPI edges between the same GWAS hits on 50,000 randomized networks (maintaining the

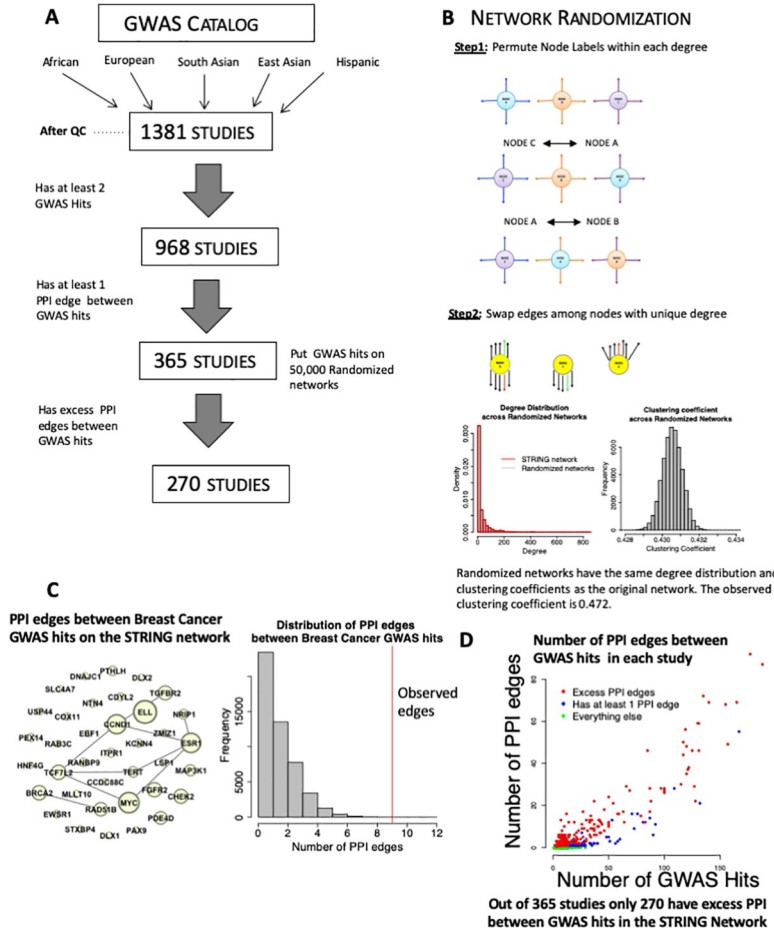

**Fig 1. Identifying studies with excess PPI amongst GWAS hits.** (A) A subset of GWAS studies had excess protein-protein interactions amongst their GWAS hits. 1,381 GWAS studies were obtained from the GWAS catalog, of these 968 had at least 2 GWAS hits and 365 studies had at least 1 PPI edge between GWAS hits. Out of these 365 studies, 270 studies had more PPI edges between their GWAS hits than expected by chance. (B) (top) Schematic of our procedure for randomizing networks. We implemented the method from [27]. To maintain topology, randomized networks were created by swapping labels among nodes with the same degree and swapping edges among nodes with unique degree. (bottom) Degree distribution and distribution of clustering coefficients across 50,000 randomized networks. (C) (left) Illustrative GWAS study (Breast Cancer risk, (GCST001937)) with 37 GWAS hits, and 9 PPI edges between GWAS hits. The size of each node denotes its degree, and edges between nodes indication PP interactions. (right) Results of permutation testing reveals that the observed 9 PPI edges between the 37 GWAS hits is in excess of the expected number of PPI edges obtained from 50,000 randomized networks. (D) In blue are the 365 GWAS studies with at least 1 PPI amongst their GWAS hits, in red are the 270 studies with more PPI amongst their GWAS hits than expected by chance (FDR < 0.05).

same degree distribution and network topology of the original STRING network, (**Fig 1B and 1C**, **S1B Fig**)). Of the 1,381 GWAS studies, 365 studies had at least 2 GWAS hits and 1 PPI edge between GWAS hits (**Fig 1A**). Of these 365 studies, 270 studies that had more PPI edges between GWAS hits than expected by chance ($p < 0.05$) (Methods) (**S2 Table**). Although PPI is related to the number of GWAS hits, studies with excess PPI were not limited to those with high numbers of GWAS hits (**Fig 1D**). Importantly, studies with excess PPI did not have a significantly different number of GWAS hits (mean = 26.196) than those without excess PPI (mean = 28.663, $p = 0.509$, t-test) (**S2 Fig**). Further, the mean degree of GWAS hits in the studies with excess PPI edges and those without was similar, 28.405 and 29.969 respectively ($p = 0.117$, t-test).

Since the majority of GWAS hits are in intergenic regions, and the putative causal gene is inferred by the closest location [29], it is likely that this inference may often not identify the true causal gene. To demonstrate this empirically, we replaced one or all of the GWAS hits with the second closest gene to the GWAS hit and compared the proportion of studies with excess PPI to our original finding of enrichment in 74% of studies. After replacing only one of the GWAS hits in each study we found 50% of studies had excess PPI between GWAS hits, while replacing all hits with the second closest gene lead to only 21% of studies with excess PPI. Finding studies with excess PPI, even after replacing all GWAS hits indicates that in at least some cases, the second closest gene may be more likely to be the causal gene.

## Detecting 'PPI Genes' in Individual GWAS studies

We next sought to identify 'PPI genes', within each GWAS study by using a hyper-geometric ratio test to find nodes within the STRING network that have excess PPI to GWAS hits given the degree and the total number of GWAS hits in the study (see **Box 1** for definitions, **Fig 2A and 2B**) (Methods). We detected 'PPI genes' in 343 of the 1,381 studies, 178 of these studies also had excess PPI among GWAS hits (**Fig 2C**, **S3 Table**). The number of 'PPI genes' detected per study ranged from 1–178 with a mean 16.227 (**S3 Fig**).

> ## Box 1: Definitions
>
> **GWAS:** *Genome-wide Association Study*
>
> **GWAS Hit:** *Gene within genome-wide significant loci ($p < 5x10^{-8}$)*
>
> **PPI:** *Protein-Protein Interaction*
>
> **Core Gene:** *Gene with direct role in disease*
>
> **Candidate Core Gene**\*: *Genes identified by our method that are likely to be core genes*
>
> **PPI gene**\*: *Gene with excess PPI with GWAS hits. Can be PPI only or PPI-GWAS. We demonstrate in the manuscript that PPI genes are candidate core genes*
>
> **PPI only:** *PPI genes that are not GWAS hits*
>
> **PPI-GWAS:** *PPI genes that are also GWAS hits*
>
> **GWAS only:** *GWAS hits that are not PPI genes*
>
> **ALL**: *All 11,049 genes within the STRING network*
>
> **Somatically Mutated PPI genes:** *Genes with excess PPI with somatically mutated cancer genes. Can be PPI only or PPI-Somatically Mutated. We demonstrate in the manuscript that PPI genes predicted from somatically mutated cancer genes are candidate core genes.*

**PPI-Somatically Mutated:** *PPI genes predicted from somatically mutated cancer genes, that are also somatically mutated cancer genes themselves*

**Somatically Mutated:** *Somatically mutated cancer genes that are not PPI genes*

*__PPI gene__ *and* __Candidate core gene__ *mean the same thing and are used interchangeably through out the text. Our method detects* __PPI genes__*, and since* __PPI genes__ *are enriched for disease relevance, Clinvar pathogenic variants and drug targets, we argue that* __PPI genes__ *are also* __candidate core genes.__

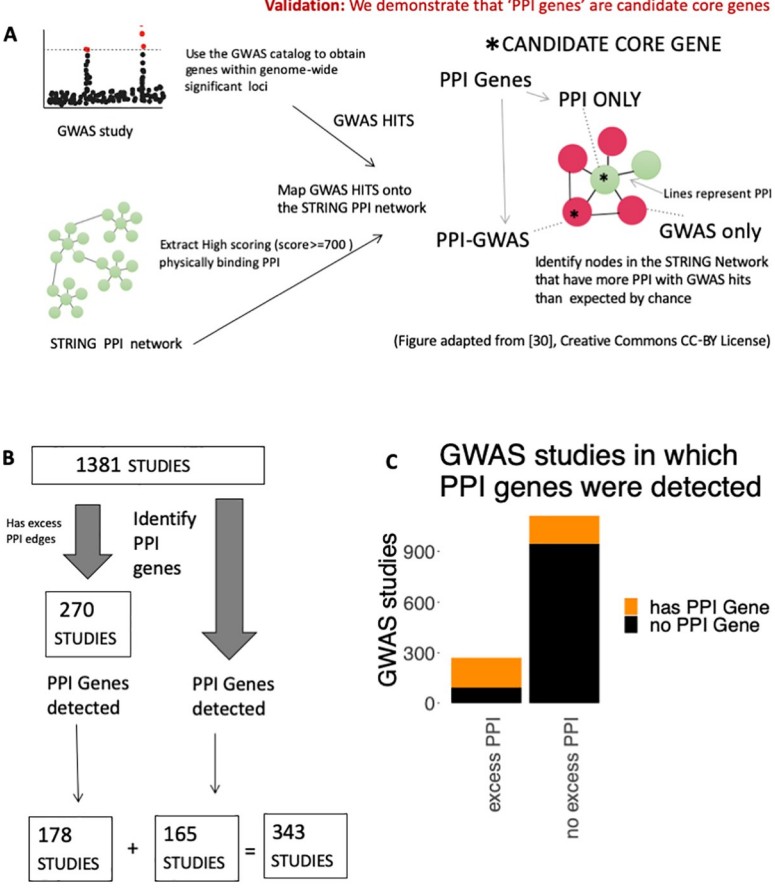

**Fig 2. Identifying 'PPI genes' using nearest neighbor GWAS hits.** (A) This figure was adapted from [30] (Creative Commons CC-BY license), it illustrates our method for detecting PPI genes. This method involves putting GWAS hits from the GWAS catalog onto the STRING PPI network, and identifying neighboring nodes that have excess PPI with the GWAS hits. We also define the following terms. 'GWAS': Genome-wide Association Study, 'GWAS HIT' Gene within genome-wide significant loci ($p < 5 \times 10^{-8}$), 'PPI': Protein-Protein Interaction. 'CORE GENE': Gene with direct roles in disease biology 'CANDIDATE CORE GENE': Gene identified by our method that is likely to be a core gene. 'PPI Gene': PPI only or PPI-GWAS gene, 'GWAS ONLY': GWAS hit that is not a PPI gene. 'PPI-GWAS': PPI gene that is also a GWAS hit. 'PPI only': PPI gene that is not a GWAS hit. 'All': all 11,049 nodes within the STRING network. (B) Schematic demonstrating how we detected PPI genes in 343 Individual GWAS studies (178 of these studies had excess PPI between GWAS hits, while 165 studies did not show excess PPI among GWAS hits). (C) Proportion of studies with excess PPI that we also detect PPI genes in. Out of 270 studies in which we detect excess PPI amongst GWAS hits, we detect PPI genes in 178 studies. Amongst the remaining 1,111 GWAS studies, we detect PPI genes in 165 studies.

**Table 1. Examples of PPI genes.**

| PPI Gene | Study accession | P-value | Rank of p-value | GWAS hit | Study with rare variant |
|---|---|---|---|---|---|
| *BRCA1* | GCST001937 | 0.026 | 2/21 | no | [31–34] |
| *INS* | GCST001213 | 0.017 | 1/1 | no | [43] |
| *APP* | GCST002245 | 0.039 | 27/28 | no | [35–37] |
| *PCSK9* | GCST000282 | 0.001 | 6/77 | no | [38,39] |
| *SNCAIP* | GCST002544 | 0.011 | 1/1 | no | [40,41] |
| *LPL* | GCST000132 | 0.001 | 7/83 | no | [44] |

Strikingly, we found that many of the PPI genes had strong disease relevance (**Table 1**). For instance, we detected *BRCA1* [31–34] in Breast Cancer (**Table 1**, **Fig 3A**, **S4 Fig**, **S3 Table**), Amyloid Precursor Protein (*APP*) in Alzheimer's Disease [35–37], *INS* in A1C measurement and Type 2 Diabetes, *PCSK9* in LDL cholesterol levels [38,39] and *SNCAIP* [40,41] in Parkinson's Disease, and the circadian rhythm gene *CRY2* in Morning vs. Evening chronotype [42].

Because our method surveys each node in the STRING network in an unbiased way, it is possible for GWAS hits to also be PPI genes themselves. To better understand differences between these classes of PPI genes, we termed 'PPI genes' that also are GWAS hits as 'PPI-GWAS', while 'PPI genes' that are not GWAS hits are 'PPI only', and GWAS hits that are not 'PPI genes' are "GWAS only" (see also **Box 1**). Surprisingly, there were many more PPI only (n = 1,821) compared to PPI-GWAS (n = 172) across the individual GWAS studies (**Fig 3B**). Suggesting that most 'PPI genes' are not detected by GWAS alone.

## Clinvar enrichment: PPI genes are candidate core genes

The omnigenic model proposes that core genes are central to disease pathogenesis, and can be detected by deep sequencing to identify deleterious, rare variants [9]. We hypothesized that if our PPI genes are enriched for Clinvar pathogenic variants, that would suggest they are good candidates for core genes as described by the omnigenic model. We looked for overlaps between our PPI genes and genes with pathogenic variants within Clinvar [45] (**Fig 3C**, **S4 Table**), and found a higher proportion of Clinvar pathogenic variants in PPI only compared to GWAS only (33% vs. 29% respectively, $p < 0.0047$, **Fig 3C**) and PPI-GWAS compared to GWAS only (49% vs. 29%, $p < 1.092 \times 10^{-7}$). We found that this enrichment remained, even after correcting for node degree (**S5 Fig**). Suggesting that the PPI genes we detect are candidate core genes.

## Enriched Disease association: PPI genes are candidate core genes

The omnigenic model also suggests that core genes are likely to be more disease associated [9]. We hypothesized that if the PPI-GWAS identify more disease relevant GWAS loci, these may have stronger GWAS p-values than GWAS hits that are not PPI genes (GWAS only). To test this, we extracted GWAS p-values for each GWAS hit from the GWAS catalog. Then compared the relative rankings of PPI-GWAS and GWAS only p-values within each study, and computed a 'rank fraction', where the p-value rank (lower p-values associated with lower ranks) is divided by the total number of GWAS hits in the study (**S5 Table**). Of the 343 GWAS studies with PPI genes, 99 contained at least 1 PPI-GWAS. Within these 99 GWAS studies we found that PPI-GWAS tended to have lower ranks and more significant p-values compared to

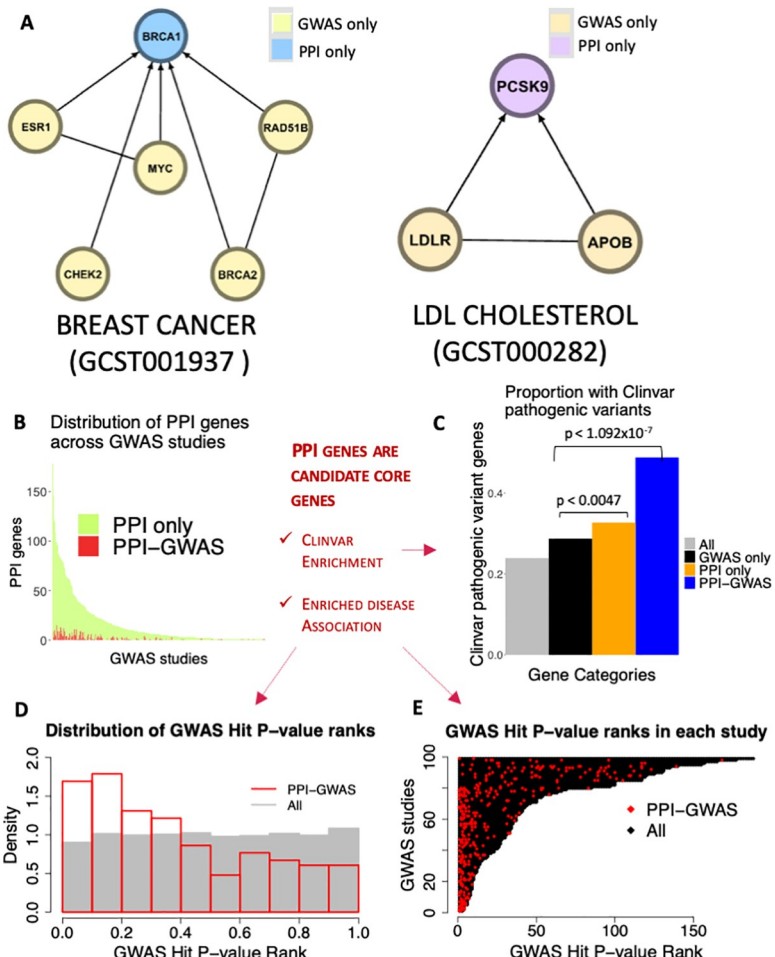

**Fig 3. PPI gene examples.** (A) Representative examples of PPI genes predicted by our method. The lines between nodes represent PPI, and the GWAS hits are depicted as pointing to PPI genes. The corresponding GWAS catalog study accessions are shown within parentheses. (B) Proportion of PPI genes within the 343 studies that we detect PPI genes in that are PPI only compared to PPI-GWAS. (C) Proportion of Clinvar pathogenic variant containing genes in All 11,049 nodes, GWAS only, PPI only and PPI-GWAS. We found enrichment in PPI only compared to GWAS only ($p < 0.0047$, Fisher's Exact Test) and in PPI-GWAS compared to GWAS only ($p < 1.092 \times 10^{-7}$, Fisher's Exact Test). (D) Out of 343 GWAS studies in which we detected PPI genes, 99 had at least 1 PPI-GWAS. We ranked the GWAS hits within each study by the GWAS p-value, and then computed rank fractions (rank/number of GWAS hits in the study). In gray is the distribution of rank fractions, across all GWAS hits from the 99 studies, in red is the distribution of rank fractions amongst only the PPI-GWAS. The PPI-GWAS distribution is skewed to the left indicating that PPI-GWAS tend to have more significant p-values compared to GWAS only. (E) In black are the ranks of all GWAS hits in the 99 GWAS studies, and in red are the ranks of only the PPI-GWAS. PPI-GWAS tend to cluster towards the left of the plot, indicating that PPI-GWAS tend to have more significant GWAS p-values.

GWAS only (**Fig 3D and 3E**). For example, 42% of PPI-GWAS were ranked in the top 25% of P-values ($p < 1.638 \times 10^{-11}$) while 17% were ranked in the top 10% ($p < 2.333 \times 10^{-5}$) (**S5 Table**). This enriched disease association of PPI-GWAS compared to GWAS only, suggests that PPI genes are candidate core genes.

## Detecting PPI genes in overarching disease processes (Parent Terms)

We subsequently sought to identify PPI genes in overarching disease processes. Conveniently, the GWAS catalog annotates related GWAS studies with a parent term. For instance, cancer

related GWAS studies are grouped under the 'Cancer' parent term, and neurologically related GWAS studies are grouped under the 'Neurological Disorders' parent term. In order to detect PPI genes in over-arching diseases processes, we merged GWAS hits from GWAS studies with the same GWAS catalog [29] provided parent term.

Similar to our analysis of individual GWAS studies, we first sought to identify parent terms with excess PPI edges between GWAS hits, and compared PPI edges in GWAS studies aggregated by parent term against a similar analysis across 50,000 randomized networks (controlled for degree distribution and network topology). We found that 16 of 17 parent terms had excess PPI between GWAS hits (**Fig 4A**), and the only parent term without excess PPI was 'Liver Enzyme Measurement'. To detect PPI genes, we applied the same hyper-geometric ratio test based method described above (Methods) to the merged GWAS hits from each parent term. Our method detected PPI genes in 12 parent terms (mean of 73.1, range: 5–255, **S6 Table**, **S3 Fig**). The parent terms we do not detect PPI genes in are 'Response to Drug', 'Other Trait', 'Metabolic Disorder' and 'Body Measurement'.

Again, we found striking disease relevance (**S6 Table**). For instance, we found *PALB2* in Cancer [46,47], *JAK-STAT* pathway in 'Digestive System Disorder' [48,49] and *LPL* (Lipoprotein Lipase) in 'Lipid or Lipoprotein Measurement' [47,50,51]. Consistent with the omnigenic prediction that core genes have deleterious rare variants, is prior analysis of UK biobank data. Which found protein truncating variants [47] in *PALB2* associated with breast cancer diagnosis and a family history of breast cancer, and protein truncating variants in *LPL* associated with decreased risk for high cholesterol [47].

Similar to our analysis of individual GWAS studies, there were many more PPI only (n = 459 unique candidate core genes) compared to PPI-GWAS (n = 194) across parent terms (**Fig 4B**). Comparison of PPI gene proportions across parent terms revealed 'Lipid or Lipoprotein Measurement' had the highest proportion (41%) of PPI-GWAS (**Fig 4B**, **S7 Table**), suggesting that lipid trait GWAS are better able to uncover PPI genes.

## Clinvar enrichment: Parent Term PPI genes are candidate core genes

Lastly, we explored the overlap between parent term PPI genes and genes containing pathogenic variants within Clinvar[45] (**Fig 4C**, **S4 Table**). We found a higher proportion of Clinvar pathogenic variants in PPI only compared to GWAS only (39% vs. 28%, $p < 8.015 \times 10^{-6}$) (**Fig 4C**), and between PPI-GWAS and GWAS only (38% vs. 28%, $p < 0.004$). Interestingly, the PPI only genes from the parent term analysis identified a stronger association with Clinvar pathogenic variants compared to the PPI only genes from the individual GWAS studies (39% vs. 33%, $p < 0.015$, **Fig 3C**), suggesting that the parent term analysis might have more power to detect PPI genes. To ensure that the Clinvar pathogenic variant containing genes that overlap with our parent term PPI genes are disease relevant, we also performed pathway enrichment analysis (Methods), and identified clearly disease relevant pathways in parent terms (**S8 Table**). Our finding of enrichment for Clinvar pathogenic variants amongst parent term PPI genes suggests that parent term PPI genes are candidate core genes.

## Enriched disease relevance: Parent Term PPI genes are candidate core genes

We next investigated the Cancer parent term PPI genes in further detail. The Cancer parent term contained 403 GWAS hits merged from 142 cancer GWAS studies from various types of cancer (**S9 Table**), and from these we identified 109 cancer parent term PPI genes (78 PPI only, 31 PPI-GWAS (**S6 Fig**)). Pathway enrichment analysis revealed that these 78 PPI only were enriched for DNA repair pathways (**S10 Table**) amongst others.

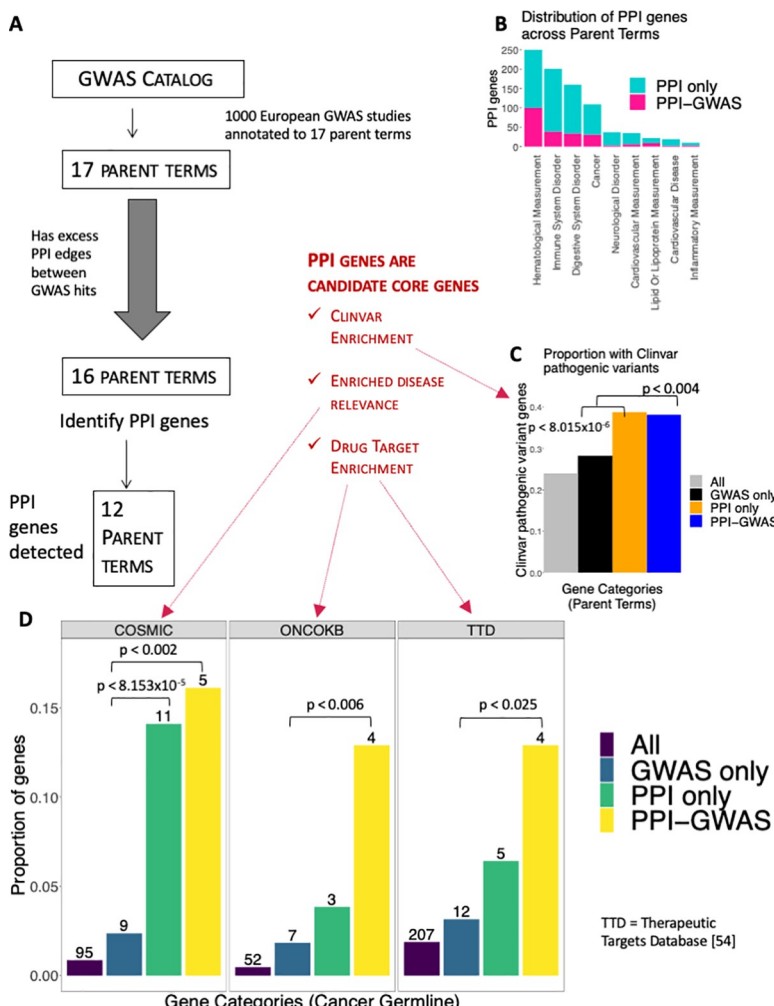

**Fig 4. PPI genes in Parent Terms.** (A) Each study in the GWAS catalog is annotated with parent terms. In order to find PPI genes amongst parent terms, we merged GWAS hits from different GWAS studies with the same parent term. We first looked for evidence of excess PPI among parent term GWAS hits. Out of 17 parent terms, we found excess PPI among GWAS hits within 16 parent terms and detected PPI genes in 12 parent terms. (B) The proportion of PPI genes in each parent term that is PPI only vs. PPI-GWAS. The 'Lipid and Lipoprotein Measurement' parent term had the largest proportion of PPI-GWAS. (We excluded the parent terms 'Biological Process', 'Other Measurement' and 'Other Disease' due to vagueness). (C) Proportion of Clinvar pathogenic variant containing genes in All 11,049 nodes, GWAS only, PPI only and PPI-GWAS. We found enrichment in PPI only compared to GWAS only (p < 8.015x10$^{-6}$, Fisher's Exact Test) and in PPI-GWAS compared to GWAS only (p < 0.004, Fisher's Exact Test). (D) Overlap of cancer parent term PPI genes with COSMIC, ONCOKB or Therapeutic Targets Database genes respectively. The numbers on top of each bar correspond to the number of PPI genes that overlaps with the genes from that gene category. COSMIC: Cancer Parent Term PPI genes were compared to the COSMIC cancer gene census germline genes [52]. Compared to GWAS only, we found that PPI-GWAS (p < 0.002, Fisher's Exact Test) and PPI-only (p < 8.153x10$^{-5}$, Fisher's Exact Test) were enriched for overlaps. ONCOKB: Cancer drug targets were obtained from Oncokb [53] and we found enrichment in PPI-GWAS (p < 0.006) compared to GWAS only. TTD: Cancer drug targets were obtained from the Therapeutic Targets Database [54] (Methods), and we found enrichment in PPI-GWAS (p < 0.025, Fisher's Exact Test) compared to GWAS only.

Since cancer involves both germline predisposition involving rare variants as well as somatic acquisition of mutations in critical genes, we sought to determine how our germline identified PPI genes compared with known cancer driver gene lists, such as the COSMIC cancer gene census [52]. Comparison of the COSMIC germline cancer gene census (n = 95) (**S11 Table**), revealed preferential enrichment for PPI only to GWAS only (14% vs. 2.4%,

$p < 8.153 \times 10^{-5}$) and PPI-GWAS to GWAS only (16% vs. 2.4%, $p < 0.002$) (**Fig 4D**). This pattern held up even after correcting for node degree (**S7 Fig**).

## Drug Target Enrichment: Parent Term PPI genes are candidate core genes

To investigate whether Cancer PPI genes are enriched for drug targets. We obtained cancer drug targets from Oncokb (n = 52) [53] (**S12 Table**), and found enrichment in PPI-GWAS compared to GWAS only (13% vs. 2%, $p < 0.006$, **Fig 4D**). For confirmation, we also performed a similar analysis using the 207 cancer drug targets obtained from the Therapeutic Target Database [54] (**S13 Table**)(Methods) and found similar results (**Fig 4D**). PPI-GWAS was enriched compared to GWAS only (13% vs. 3.1%, $p < 0.025$). Importantly these patterns held up even after correcting for node degree (**S7 Fig**).

## Detecting PPI genes from Somatically Mutated cancer genes

We hypothesized that our method may also work on disease-associated genes obtained from methods other than GWAS, such as genes enriched for somatic mutations in tumors. We obtained genes enriched for somatic mutations across 21 tumor types [55] and after filtering for those within our list of 11,049 genes (**S1 Table**), found 218 somatically mutated cancer genes (**S14 Table**).

To determine whether these 218 somatically mutated cancer genes have more PPI edges with each other than expected by chance, we counted the number of PPI edges between the 218 somatically mutated cancer genes on the STRING network and compared this to the number of PPI edges between the same somatically mutated cancer genes across 50,000 randomized networks (maintaining degree distribution and network topology). In the STRING network we observed 495 PPI edges between the 218 somatically mutated cancer genes while the largest number of PPI edges between the same 218 somatically cancer genes in any of the other randomized networks was only 366, suggesting that somatically mutated cancer genes have more PPI with each other than expected by chance (**Fig 5A**, **S8 Fig**).

To detect PPI genes from these 218 somatically mutated cancer genes, we implemented the same method described above, but instead of GWAS hits we used 218 somatically mutated cancer genes (**S14 Table**), and detected 843 PPI genes (**S15 Table**) (we refer to these as somatically mutated PPI genes, see **Box 1**). When we compared these 843 somatically mutated PPI genes to the 218 somatically mutated cancer genes themselves, we found an overlap of 101 (we refer to these as 'PPI-Somatically Mutated'), which means that 742 of the somatically mutated PPI genes represent novel associations (we refer to these as 'PPI only', see **Box 1**, **S15 Table**). Indicating that similar to the individual GWAS studies (**Fig 3B**, **S3 Table**) and parent term analysis (**Fig 4B**, **S6 Table**), there are many more PPI only compared to PPI-Somatically Mutated (**Fig 5B**). Intriguingly out of 843 somatically mutated candidate core genes (**S15 Table**) ranked second with $p < 1.3 \times 10^{-11}$ was *TP53*, *TP53* has a degree of 284 and was found to have PPI with 31 of the 218 somatically mutated genes (**Fig 5C**, **S15 Table**).

We also compared degree distributions and found that PPI genes tended to have higher degree (**S9 Fig**).To better understand the relationship between node degree and number of PPI edges with the 218 somatically mutated cancer genes, we plotted the number of PPI edges between every node in the network and the node degree (**Fig 5D**), and found that although higher degree nodes have more PPI edges, PPI genes have more PPI edges than expected given their degree. To further illustrate that the PPI genes signal is not driven by node degree, we also computed the proportion of somatically mutated PPI genes within degree bins (**S10 Fig**), and found little evidence that PPI genes were driven by higher degree nodes.

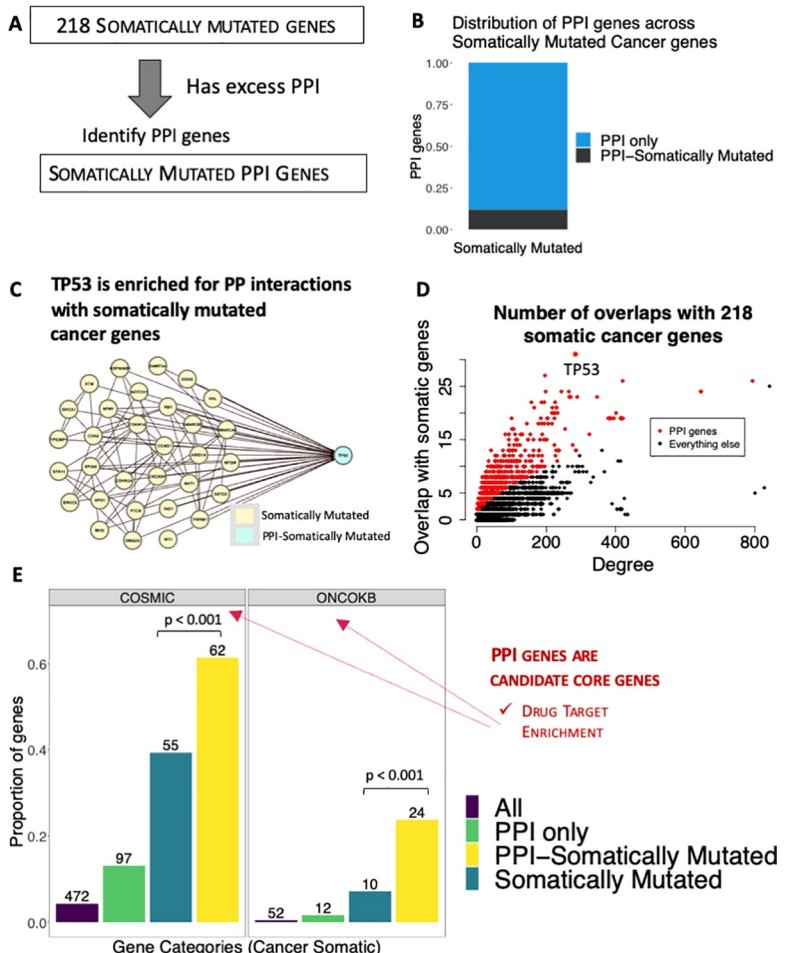

**Fig 5. PPI Genes identified from Somatically mutated Cancer Genes.** (A) Schematic showing how we assessed 218 somatically mutated cancer genes for excess PPI, and then used them to predict PPI genes. 'PPI-Somatically Mutated': PPI genes predicted from somatically mutated cancer genes, that are also somatically mutated cancer genes themselves. 'Somatically Mutated': Somatically mutated genes that are not PPI genes. 'PPI only': PPI genes that are not in the list of 218 somatically mutated genes 'All': all 11,049 nodes in the network. (B) We detected 843 PPI genes from the somatically mutated cancer genes, 742 were PPI only while 101 were PPI-Somatically Mutated. (C) Amongst the PPI genes we predicted from the somatically mutated genes, ranked 2nd was *TP53*. *TP53* has a degree of 284 and has PPI with 31 of the 218 somatically mutated genes. *TP53* is shown in cyan, and in yellow are the 31 somatically mutated cancer genes that have PPI with *TP53*. The PPI edges are represented as lines. (D) To better understand the relationship between node degree and the number of PPI edges to the 218 somatically mutated genes, we took each node in the network (n = 11,049) and counted the number of PPI edges to the 218 somatically mutated cancer genes. We plotted the number of PPI edges of each node compared to the degree of the node and show PPI genes in red, and highlight *TP53* with an asterisk. (E) Overlap of somatically mutated PPI genes with COSMIC, ONCOKB or Therapeutic Targets Database genes respectively. The numbers on top of each bar correspond to the number of PPI genes that overlaps with the genes from that gene category. COSMIC: Somatically mutated genes and PPI genes were split into PPI-only, PPI-Somatically Mutated, Somatically Mutated and All (see **Box 1** for definitions). Then each category was compared to the COSMIC cancer gene census somatic genes [52]. We found that PPI-Somatically mutated (p < 0.001, Fisher's Exact Test) was enriched compared to Somatically mutated. ONCOKB: Comparison to cancer drug targets from Oncokb [53], revealed PPI-Somatically mutated was enriched compared to Somatically mutated (p < 0.001, Fisher's Exact Test).

## Enriched disease relevance: Somatically Mutated PPI genes are candidate core genes

Interestingly our method detected *TP53* to be ranked second amongst all somatically mutated PPI genes. *TP53* was in our list of 'PPI-Somatically Mutated' (**S15 Table**), because in addition

to being a PPI gene, it was also a somatically mutated cancer gene itself (**S14 Table**). This finding suggests that in addition to being somatically mutated in cancer, *TP53* may also act like a "hub" or "master-regulator" for other Somatically Mutated genes. Ranked first amongst the somatically mutated PPI genes was *CTNBB1* ($p < 1.12 \times 10^{-15}$) which is also well known in cancer pathogenesis (**S15 Table**). The strong disease relevance of *TP53* and *CTNBB1* within the context of cancer, suggests that the PPI genes we detect from the somatically mutated cancer genes are candidate core genes.

To further assess disease relevance, we compared the COSMIC cancer gene census [52] somatic genes (n = 472) (**S16 Table**) to our candidate core genes from the somatically mutated analysis, and found enrichment amongst PPI-Somatically Mutated compared to Somatically Mutated (61% vs. 39%, $p < 0.001$) (**Fig 5E**). This pattern remained even after node degree correction (**S11 Fig**).

## Drug Target Enrichment: Somatically Mutated PPI genes are candidate core genes

We also compared the PPI genes we detected from the somatically mutated analysis to Oncokb [53] drug targets (n = 52) (**S12 Table**) and found that PPI-Somatically Mutated was enriched compared to Somatically Mutated (24% vs. 7% $p < 0.001$) (**Fig 5E**), even after correcting for node degree (**S11 Fig**), suggesting that the somatically mutated PPI genes we detect are candidate core genes.

## Comparison across different interaction types in STRING Network

The method we have proposed here, could in theory be applied to any network. We've applied it to the STRING network's physically binding (mode = "binding") PP interaction network due to prior evidence that GWAS hits physically interact with each other more than expected by chance [27]. However, there are 6 other interaction types captured within the STRING network (mode = "reaction/catalysis/activation/inhibition/ptmod(post-translational modification)/expression"), and the STRING network contains a number of experimental assays, one of which is coexpression. In order to assess the performance of our method across different interaction types, we built PPI networks on the 6 other interaction types and for co-expression(Methods).

In order to facilitate comparison, we used the 403 cancer parent term GWAS hits to detect PPI genes across the 8 networks (**S17 Table**), and filtered each network for the 11,049 nodes from our physically binding network. To assess performance, we counted the number of PPI genes, checked whether *TP53* was detected, and whether there was enrichment for Clinvar pathogenic variants and for cancer drug targets from Oncokb [53]. Comparison across 8 different networks revealed that the physically binding network (mode = "binding") described in this manuscript outperformed the other networks (**Table 2**), however this might be due to this network having more nodes.

**Table 2. Comparison of our method applied to networks built with different interaction types.**

| Type | # nodes | # PPI genes | Is *TP53* a PPI gene? | Clinvar | Oncokb |
|------|---------|-------------|----------------------|---------|--------|
| Reaction | 5583 | 116 | no | | |
| Binding | 11,049 | 109 | yes | enriched | enriched |
| Catalysis | 5,277 | 21 | no | | |
| Activation | 5,229 | 35 | yes | enriched | |
| Inhibition | 2,802 | 3 | no | | |
| Post-translational Modification | 2,566 | 8 | no | | enriched |
| Expression | 1,460 | 55 | no | | enriched |
| Co-expression | 8,231 | 167 | yes | | |

## Discussion

By combining GWAS hits with PPI networks, we have developed a method for detecting 'PPI genes', and after detailed characterization found that these PPI genes are enriched for disease relevance, Clinvar pathogenic variants and drug targets–suggesting they are candidate 'core genes' as defined by the omnigenic model [9]. We applied our method to 1,381 GWAS studies, and detected candidate core genes in 343 individual GWAS studies, 12 overarching parent terms, and somatically mutated cancer genes.

We demonstrate that GWAS hits that are also candidate core genes (i.e. PPI-GWAS), are more strongly statistically associated with the underlying trait than those loci that are not candidate core genes (i.e. GWAS only). In addition, we demonstrate that many candidate core genes have no excess GWAS signal (PPI only)—suggesting that they are unlikely to be detected in GWAS of larger sample size. We present a new application of GWAS data: identifying core genes as envisioned by the omnigenic model that do not themselves exhibit GWAS signal by consulting physical interactions. Our results provide unique insights into disease biology and suggest that GWAS can be combined with PPI networks to detect "core genes".

Our method was able to uncover novel insights into cancer, these include detecting 109 candidate core genes from germline GWAS hits and 843 candidate core genes from genes enriched for somatic mutations in tumors [55]. We also found that somatically mutated cancer genes are connected to each other with excess PPI, suggesting an underlying biological process such as a protein complex unifying the seemingly disparate somatically mutated cancer genes. In addition, we found *TP53* to be a highly statistically significant PPI-Somatically Mutated candidate core gene, suggesting that in addition to being somatically mutated itself, it may also be a regulator of other somatically mutated cancer genes.

Although the omnigenic model is usually described in terms of gene-regulatory networks [9], it is just as plausible for peripheral gene effects to be mediated by PPI as it is for them to be mediated by gene-regulatory networks. An important implication of our work, is that it suggests that GWAS hits are more important, than is currently captured by the omnigenic model [9]. Adding an additional tier comprising of GWAS hits, such that core genes are at the center, followed by GWAS hits, then peripheral genes expressed in the right cell type, followed by all other genes might better capture the importance of GWAS hits.

Although our data and results are compelling, some caveats and limitations remain. Firstly, we obtained GWAS hits from the GWAS catalog [29], rather than summary statistics, which could introduce heterogeneity in terms of methods and sample sizes used by different research groups. However, we attempted to mitigate this by only considering loci within the stringent genome-wide significance p-value threshold ($5 \times 10^{-8}$). Another potential limitation is that we assume GWAS hits are correctly associated with causal variants, and that non-coding variants, are linked to the correct gene. The GWAS catalog [29] lists genes reported in the literature, and also uses an automated pipeline to map the gene closest to the most associated SNP in each loci. This is in line with what is known about causal genes [27], however it's difficult to know whether the annotated genes are causal. We tested this empirically and found that although the second closest gene is likely to be causal in some cases, the proportion of studies with excess PPI is highest with the first closest gene. In addition, to further mitigate this risk, we limited our analysis to loci annotated with only 1 or 2 genes within the GWAS catalog. Another limitation might be that the Clinvar enrichment at PPI-GWAS compared to GWAS only, could be due to the PPI-GWAS gene being the actual causal rather than the GWAS only gene. Although it is difficult to rule out this possibility, our finding that replacing GWAS hits with the second closest gene results in a lower proportion of studies with excess PPI edges, indicates that the GWAS hits are likely to be the causal gene in a number of studies.

In our parent term analysis we attempted to predict candidate core genes by combining studies with related traits. However, these studies may not be independent, in addition, the different disease studies may not be equally represented. For instance, in the cancer parent term, breast and prostate cancer studies were more highly represented (**S9 Table**), suggesting that the cancer core genes may be biased towards these cancer types.

We suggest that, future work may involve using other methods to validate our candidate core gene predictions, using deep sequencing to identify rare, deleterious variants and wet lab experiments to validate disease relevance. In addition, applying our methods to tissue and cell-type specific PPI networks is likely to increase accuracy, by leveraging disease specific PP interactions that may not be present in the current network.

We have presented here, a simple, yet powerful method for detecting candidate core genes, by intersecting GWAS hits with human protein-protein interactions. The significance of this, is that even if most core genes cannot be detected by GWAS directly, intersecting with predefined PPI networks enables GWAS hits to be used as a hook to uncover core genes in a 'guilt by protein interaction' approach. Our approach adds another layer to interpreting GWAS signal.

## Methods

### Protein-protein interaction network

We downloaded the files 9606.protein.links.v10.5.txt, 9606.protein.actions.v10.5.txt and protein.aliases.v10.5.txt from version 10.5 of the STRING PPI network [56] (https://STRING-db. org/cgi/download.pl?sessionId=x8qiRTBCS4ps). The 9606.protein.links.v10.5.txt file contains scored links between proteins, the 9606.protein.actions.v10.5.txt file contains interaction types and the protein.aliases.v10.5.txt contains aliases for STRING proteins.

In order to maintain standard nomenclature for gene names, we also downloaded the HGNC gene database, by navigating to the Downloads page from the HGNC website https:// www.genenames.org/cgi-bin/statistics, and downloading the text file, which contained 19,198 protein products per loci group. We used custom scripts to convert the Ensembl gene ids from STRING, into gene aliases (using protein.aliases.v10.5.txt), and then converted these aliases to current approved gene HGNC symbols. We used the 9606.protein.links.v10.5.txt file to extract the protein links, and in order to filter for physically binding PP interactions, we then navigated to the file 9606.protein.actions.v10.5.txt, and used the third column entitled 'mode' and filtered for entries containing the term 'binding'. We know these refer to physical PP interactions as the FAQ section of the STRING database website (http://version10.STRING-db.org/ help/faq/), states "*In order to get the physical interactions you need to download proteins. actions.(version).txt.gz from download section. If the interaction is marked as "binding" you can be sure that this is a physical interactions. If interaction does not have "binding" specified (i.e., anything else) it may be either physical or functional*". After filtering for physically binding, we then used the combined_score column of 9606.protein.links.v10.5.txt to filter for interactions with score $>= 700$. This left us with a total of 11,049 proteins within the PPI network (**S1 Table**).

### GWAS catalog

We downloaded the GWAS catalog [57] by downloading the file GWAS_catalog_v1.0.2-associations_e93_r2018-08-28.tsv from https://www.ebi.ac.uk/GWAS/downloads. The GWAS catalog [57] contains SNPs associated with diseases and traits collated from the literature. Each SNP is annotated with the closest gene, gathered either from the publication (Reported) or via an automated mapping pipeline implemented within the GWAS catalog (Mapped). The

catalog has also been annotated with parent terms, which we obtained by downloading the file gwas-efo-trait-mappings.tsv from ftp://ftp.ebi.ac.uk/pub/databases/gwas/releases/latest/. This file maps each of the GWAS studies to Parent Terms by using the shared column of disease traits. We then applied a number of QC filtering cut-offs. We found that some publications were associated with multiple genotyping arrays sometimes with different numbers of SNPS passing QC (captured by the column "PLATFORM (SNPS PASSING QC)"). To ensure we include only 1 genotyping array per publication, we filtered the column "PLATFORM (SNPS PASSING QC)" for the entry with the largest number of SNPS that pass QC in each publication. We found that some loci were associated with multiple p-values, due to different subsets of samples. In order to retain only 1 p-value for each loci, we required that the "P-VALUE (TEXT)" column was an empty string. The "P-VALUE (TEXT)" column contains information describing the context of the p-value (e.g. females, smokers) (https://www.ebi.ac.uk/gwas/docs/fileheaders). For instance, if the p-value was computed only in females the entry in the field will be "females", and if the p-value was computed only in smokers the entry in this field will contain "smokers". By requiring that the "P-VALUE (TEXT)" column was an empty STRING, we are only retaining p-values corresponding to the whole dataset.

To extract GWAS hits, we used the "mapped" and "reported" columns to extract out the mapped and reported genes that were also part of the list of 11,049 genes from the PPI network [56]. A small subset of loci had SNP rsids with "x "or ";", which we removed from the analysis. We found some instances where the same gene was linked to multiple loci. This occurs when the gene is very large and multiple GWAS signals are found across the span of the gene. However, this could be problematic in our analyses because multiple p-values will be assigned to the same gene. In order to avoid this problem, we compared the p-values from the different loci containing the same gene and assigned the lowest p-value to that gene. In order to ensure all loci are genome-wide significant, we removed loci where the p-value of the most associated SNP was above the standard genome-wide significance threshold of $5 \times 10^{-8}$. We noticed that the GWAS catalog contained some loci that contained many genes. Since it is difficult to determine which of these genes is likely to be causal, we only included loci that were annotated with 2 genes at the most.

The catalog also contains ancestry annotations, which we obtained by downloading the file GWAS_catalog-ancestry_r2018-08-28.tsv. In order to separate out studies based on Ancestry; we used the "BROAD ANCESTRAL CATEGORY" from the file: gwas_catalog-ancestry_r2018-08-28.tsv. We only included 5 ancestry categories (European, African American or Afro-Caribbean, Hispanic or Latin American, South Asian and East Asian). After this filtering we were left with 1,381 GWAS studies.

## Network randomization

We performed network randomization in the same manner as [27]. This method maintains the degree distribution and topology of the network as measured by clustering coefficient. The randomization was done by separating out nodes with the same degree, from nodes with unique degree. We swapped node labels amongst nodes with the same degree, we did this once for each degree, and then repeated it 1000 times. Out of 11,049 nodes, 57 nodes had unique degree, and out of 163,181 edges, 19,392 edges belonged to nodes of unique degree. We grouped nodes with unique degree and performed edge swaps, between different nodes. We performed multiple runs to optimize the number of switching between edges, and found that even after 10 million swaps 3,500 out of 19,392 edges couldn't be randomized. We settled on 100,000 edge swaps for each network.

Since the randomization was performed in 2 stages, we found a small number of edges (100) overlapped between the two steps, and were only counted once in the final network,

which led to a small number of nodes differing from the original degree distribution by 1 or 2 nodes. However, when we plotted the degree distribution of the 50,000 randomized networks in gray over-laid with the observed degree distribution of the STRING network in red (**Fig 1B**), we found that the degree distribution in the randomized networks matched that of the real STRING network.

To ensure that the topology between the randomized networks is the same as the real network, we computed clustering coefficients. The observed clustering coefficient was 0.47, while the mean clustering coefficient for our 50,000 randomized networks was 0.43 (**Fig 1B**). A similar difference was observed in [27], which is the where we obtained our method from.

## Identifying studies enriched for PP interactions amongst GWAS hits

To identify studies enriched for PP interactions among GWAS hits, we used the STRING network to count the number of PPI edges. Out of 1,381 studies, we found that 365 studies had at least 2 GWAS hits with at least 1 PPI edge between them. We took the GWAS hits from these 365 studies and counted the PPI edges between them in each of the 50,000 randomized networks. We then counted the number of randomized networks that had PPI edges greater than or equal to the number of PPI edges observed between the GWAS hits in the real STRING network. If the number of PPI edges observed between the GWAS hits in the real STRING network is greater than the number of PPI edges between the same GWAS hits in $> 95\%$ of randomized networks, it was deemed to have more PPI edges than expected by chance.

## Replacing GWAS hits with the second closest hit

We created a bed file with locations of the GWAS hits, then used bedtools [58] 'closest' feature to get the closest gene. In order to maintain the number of GWAS hits in the study, if the first closest gene was already in the list we got the next closest gene. After replacing the GWAS hits with the second closest GWAS hit, we put the GWAS hits on the 50,000 randomized networks to identify studies with excess PPI edges ($p < 0.05$). We also counted the number of studies with at least 1 PPI edge between GWAS hits, and after replacing with the second closest GWAS hit, we calculated the proportion of studies with at least 1 PPI edge that have excess PPI edges.

## PPI gene detection

We implemented our method for detecting PPI genes using custom Perl scripts. For each GWAS study we extracted out all the GWAS hits and then compared the GWAS hits to the directly interacting proteins of each protein in the network (**S1 Table**). We then counted the overlap between the GWAS hits for each study and the directly interacting proteins at each node in the network. We then extracted out 3 numbers for every protein node that overlaps with at least 1 GWAS hit. These numbers are: (1) the number of GWAS hits per study (2) the number of directly interacting proteins that are also GWAS hits (3) the degree of the node. We then used custom R scripts to perform a hyper-geometric ratio test to assess whether the level of overlap between GWAS hits and directly interacting proteins is enriched given the total number of GWAS hits in the study, and the degree of the node. We then used the number of nodes that overlap with at least 1 GWAS hit to perform Benjamini-Hochberg multiple testing correction of the raw p-values.

## Filter out PPI genes detected by less than 2 GWAS hits

We then counted the number of GWAS hits that detect each PPI gene and in order to reduce spurious signals, we removed PPI genes that only directly interact with 1 GWAS hit.

## Filtering out PPI genes detected by GWAS hits from the same locus

In order to ensure that the GWAS hits that detect each PPI gene are from 2 different loci, we computed the genomic distance between GWAS hits that detect each PPI gene, and removed PPI genes that are only detected by GWAS hits within 1MB of each other.

## Case study: PPI gene Detection in Breast cancer study (GCST001937)

Here we describe in detail how we detected PPI genes in the breast cancer study (GCST001937). The breast cancer study GCST001937 has 37 GWAS hits extracted from the mapped and reported columns of the GWAS catalog that are also within the 11,049 nodes from our STRING PPI network (**S2 Table**). We compared these 37 GWAS hits to the directly interacting proteins of each of the 11,049 nodes. We found that out of 11,049 nodes only 805 nodes have at least 1 direct PP interaction with at least 1 of the 37 GWAS hits. We computed a hyper-geometric ratio test at each of these 805 nodes, using the degree, number of overlaps between directly interacting proteins and GWAS hits and number of GWAS hits in the study (37) as input. Then used the number of nodes with at least 1 PP interaction (n = 805) for Benjamini-Hochberg correction. For example, *BRCA1* has a degree of 152 and it overlaps with 5 of the 37 GWAS hits, and after computing a hyper-geometric ratio test has a raw p-value ($p < 0.00014$), and after Benjamini-Hochberg correction it has a $p < 0.026$. We used a Benjamini Hochberg corrected p-value cut-off of 0.05, to identify PPI genes. In order to reduce spurious signals, we then removed PPI genes that only overlap with 1 GWAS hit (see "Filter out PPI genes detected by less than 2 GWAS hits" above). In order to ensure that the GWAS hits were not all coming from the same loci, we then computed the distance between the breast cancer GWAS hits and removed PPI genes that were detected by GWAS hits that are all within 1MB of each other (see "Filtering out PPI genes detected by GWAS hits from the same locus" above).

## Degree distribution of PPI genes versus the rest of the network–Violin plot

We extracted the degree of the PPI genes and compared them to the degree of the rest of the network (**S9 Fig**), and plotted these distributions on a violin plot and performed a Kolmogorov–Smirnov test to determine if the 2 distributions are different.

## Visualization of disease networks

We used custom Perl scripts and the software Gephi version 0.9.2 [59], to visualize networks using the force directed 'Fruchterman Reingold' layout algorithm [60].

## Parent term analysis

We used GWAS catalog provided parent term annotations to merge GWAS hits with the same parent term among 1000 European Ancestry GWAS studies (**S2 Table**).

## Parent terms enriched PPI between GWAS hits

In order to identify parent terms whose GWAS hits have excess PPI edges, we counted the number of PPI edges between the GWAS hits on the STRING network, and the number of PPI edges between GWAS hits on 50,000 randomized networks. Parent terms containing PPI edges between GWAS hits observed in less than 5% of randomized networks, were deemed to have more PPI between GWAS hits than expected by chance.

## PPI gene detection in Parent Terms

We compared the merged GWAS hits in each parent term to the directly interacting proteins of each node within the STRING network, to compute the overlap between GWAS hits and directly interacting proteins. We extracted out 3 numbers for every protein node that overlaps with at least 1 GWAS hit: (1) the number of GWAS hits per study (2) the number of directly interacting proteins that are also GWAS hits (3) the degree of the node. We then used custom R scripts to perform a hyper-geometric ratio test to assess whether the level of overlap between GWAS hits and directly interacting proteins is enriched given the total number of GWAS hits in the study, and the degree of the node. We then used the number of nodes that overlap with at least 1 GWAS hit to perform Benjamini-Hochberg multiple testing correction of the raw p-values. Nodes with Benjamini Hochberg corrected p-values < 0.05 were deemed parent term PPI genes. Similar to the PPI gene detection method for individual GWAS studies, in order to reduce spurious signal, we also filter out PPI genes detected by less than 2 GWAS hits (see "Filter out PPI genes detected by less than 2 GWAS hits" above), and filter out PPI genes detected by GWAS hits that are all from the same locus within 1MB of each other (see "Filtering out PPI genes detected by GWAS hits from the same locus" above).

## Classifying PPI genes as PPI-GWAS or PPI only

PPI genes that are also GWAS hits are PPI-GWAS while the rest are PPI only. Any GWAS hits without a PPI gene are GWAS only.

## PPI-GWAS P-value ranks

Out of all GWAS studies we detected PPI genes in that have at least 1 PPI-GWAS, we extracted the GWAS catalog provided p-values for all GWAS hits in the study. We then compared the rank of the p-value of each GWAS hit to the total number of GWAS hits in the study to compute a rank fraction (rank/number of GWAS hits in the study). Then compared the rank fraction distribution of all GWAS hits to the rank fraction distribution of PPI-GWAS. To compute enrichment, we counted the number of PPI-GWAS with p-values ranked within the top 10% and top 25% and compared this to the number of total GWAS hits ranked within the top 10% and top 25%, and used Fisher's Exact Test to compute statistical significance.

## Clinvar

We used the methods outlined in the blog https://davetang.org/muse/2017/01/30/exac-allele-frequency-pathogenic-Clinvar-variants/. Our aim was to extract out pathogenic Clinvar variants. We downloaded the Clinvar data with the file: http://ftp.ncbi.nlm.nih.gov/pub/Clinvar/vcf_GRCh37/Clinvar_20180603.vcf.gz, and found that some of the Clinvar variants contained multiple annotations, so we wrote a custom Perl script to extract out only the Clinvar variants with a single annotation. Then from these we extracted out only the Clinvar variants that were annotated as pathogenic (**S4 Table**), and are also within our network (**S1 Table**).

## Overlap between Clinvar pathogenic variant containing genes and PPI genes

We wrote custom Perl scripts to count the number of PPI genes that also contain Clinvar pathogenic variants. We then compared the proportions of PPI genes in PPI only, PPI-GWAS, GWAS only and all 11,049 proteins in the network. We did this for both the PPI genes detected from the individual GWAS studies and the PPI genes detected from the 12 parent terms. We computed statistical significance using a Fisher's Exact Test.

### Pathway enrichment of PPI genes containing Clinvar pathogenic variants

For each parent term, we extracted out the genes with Clinvar pathogenic variants, then compared these to all genes (n = 11,049) to see if any pathways were enriched. We used the Msigdb 6.2 pathway database [61] and applied Fisher's Exact Test to compute statistical significance and applied Benjamini-Hochberg multiple testing correction.

### Oncokb

We navigated to the website (https://www.oncokb.org/) and downloaded the genes from levels 1–4, and then filtered these genes for those that are within our list of 11,049 nodes. We then compared the proportion of Oncokb genes in cancer parent term GWAS hits and PPI genes and somatically mutated genes and PPI genes and also made comparisons to all the nodes in the network (n = 11,049), and computed statistical significance using a Fisher's Exact Test.

### Therapeutic Targets Database

We navigated to the website (http://db.idrblab.net/ttd/full-data-download), then navigated to 'Target to disease mapping with ICD identifiers', and parsed the downloaded file for disease entries that contain 'cancer', 'tumor' or 'Metastatic'. Then we used the uniprot ID mapping feature (https://www.uniprot.org/uploadlists/), to map the uniprot ids to gene names, then we filtered these genes for those that are within our list of 11,049 genes, and were left with 207 genes. We then compared these 207 genes to the PPI only, PPI-GWAS, GWAS only lists within the cancer parent term and PPI only, PPI-Somatically mutated, and Somatically Mutated lists, and used a Fisher's Exact Test to assess significance.

### Cosmic

We navigated to the website (https://cancer.sanger.ac.uk/cosmic) and navigated to 'Cancer Gene Census' [52] and extracted only tier1 genes, then separated these out into Germline and Somatic, we then filtered these for those within our list of 11,049 nodes. We compared these genes to Cancer Parent Term GWAS hits and PPI genes, and Somatically Mutated genes and Somatically mutated PPI genes and assessed enrichment with a Fisher's Exact Test.

### Clustering of Cancer Parent Term PPI Genes

We used the STRING network to extract the directly interacting partners of each of the 109 cancer parent term PPI genes. Then we did a pairwise comparison to count the number of directly interacting partners that are shared across each node pair, and then we used the number of shared interacting partners to weight the edge between the 2 nodes. We then used the Fruchterman Reingold force directed layout algorithm [60] within Gephi [59] to visualize the network (**S6 Fig**).

### Pathways enriched in the Cancer Parent Term PPI Genes

We compared these 78 PPI only cancer parent term PPI genes to the total list of 11,049 nodes in the network, and estimated enriched pathways using the Msigdb pathway database version 6.2 [61] which contains >17,000 pathways. We assessed enrichment with a Fisher's Exact Test, and used the Benjamini-Hochberg procedure to correct for multiple testing.

### Assessing whether somatically mutated cancer genes are enriched for PPI

We counted the number of PPI edges between the 218 somatically mutated cancer genes on the STRING network, then compared this to the number of PPI edges between the same 218

somatically mutated cancer genes on 50,000 randomized networks. In order to compute a p-value we computed what proportion of the 50,000 randomized networks, have more PPI edges between the 218 somatically mutated cancer genes than is observed in the real STRING network.

| | | |
|---|---|---|
| 218 somatic genes -> | on STRING network -> | # of PPI between somatic genes = 495 |
| 218 somatic genes -> | Randomized network1 -> | # of PPI between somatic genes = 278 |
| 218 somatic genes -> | Randomized network2-> | # of PPI between somatic genes = 300 |
| | Randomized network3 | |
| | Randomized network4 | |
| | Randomized network5 | |
| | Randomized network6 | |
| | | |
| | Randomized Network 50,000 | |

## Detecting PPI genes from somatically mutated cancer genes

To calculate PPI genes, we used the same hyper-geometric ratio test based method as above, but instead of GWAS hits, we used the 218 somatically mutated cancer genes. We compared all 218 somatically mutated cancer genes to each node in the STRING PPI network, and counted the number of directly interacting proteins at each node that overlap with the 218 somatically mutated cancer genes. We assessed enrichment by computing a hyper-geometric ratio test at each node using (1) the number of somatically mutated cancer genes that overlap with the direct PPI interactions (2) the degree of the node (3) and the total number of somatically mutated cancer genes (n = 218). We used the nodes with at least 1 PPI interaction with the 218 somatically mutated cancer genes to do Benjamini-Hochberg multiple testing correction, and deemed nodes with adjusted p-value < 0.05 to be somatically mutated PPI genes. Similar to the core gene detection method for individual GWAS studies, in order to reduce spurious signal, we also filtered out PPI genes detected by less than 2 somatically mutated cancer genes (see "Filter out PPI genes detected by less than 2 GWAS hits" above), and filtered out PPI genes detected by somatically mutated cancer genes that are all from the same locus(see "Filtering out PPI genes detected by GWAS hits from the same locus" above).

## Comparison of different interaction types

In this manuscript we only considered physically binding protein interactions, from the STRING network, we did this by requiring that the 'mode' column of the 9606.protein.actions.v10.5.txt was set to 'binding'. However, the STRING network contains other interaction types. We extracted the other interaction types setting the "mode" field to 'reaction', 'catalysis', 'activation', 'inhibition', 'ptmod' (post translational modification) and 'expression'. We then built networks using the entries from these other interaction types. We also built a co-expression network by extracting out the 10th column named 'coexpression' from the file 9606.protein.links.full.v11.0.txt, however due to limited number of nodes we did not filter the co-expression nodes by score.

In order to facilitate comparison, we filtered the nodes from the 8 different interaction networks for the11,049 nodes within our "binding" network, and then detected PPI genes using the 403 GWAS hits from the Cancer parent term (European ancestry, see S2 Table). We then compared across different networks by assessing whether *TP53* was detected as a PPI gene, whether there was enrichment for Clinvar pathogenic variants and whether there was enrichment for Oncokb drug targets.

## Supporting information

**S1 Fig. Distribution of scores across STRING network.** (A) This figure shows the distribution of scores within the STRING PPI network. The scores ranged from 150–999, with a mean of 277.65. We restricted our analyses to PPI interactions with score $> = 700$, representing 6% of all PP interactions in the network. The red vertical line represents the score cut-off of 700 we chose for our analysis. (B) Schematic showing that the observed number of PPI among GWAS hits in the STRING network tends to be higher than the number of PPI among GWAS hits in randomized networks.
(TIF)

**S2 Fig. Relationship between number of GWAS hits and excess PPI.** (A) Distribution of GWAS hits amongst studies with at least 1 PPI edge on STRING, and studies with no PPI edge between GWAS hits. (B) The proportion of studies that have at least 1 PPI edge at each number of GWAS hits. This plot demonstrates that at low GWAS hits the likelihood of having at least 1 PPI edge between the GWAS hits is low, but as the number of GWAS hits increases the chances of having at least 1 PPI edge between the GWAS hits increases. (C) Distribution of GWAS hits amongst studies with excess PPI edges between GWAS hits and amongst studies with no excess in PPI edges between GWAS hits. (D) The proportion of studies with excess PPI edges at each number of GWAS hits. Although there is a tendency for studies with low GWAS hits to have no excess in PPI edges, there are many studies with high numbers of GWAS hits that do not have excess PPI edges. (E) In gray is the degree distribution of the whole STRING network, and in red is the degree distribution of GWAS hits within GWAS studies with excess PPI (n = 270).
(TIF)

**S3 Fig. Distributions of GWAS hits and PPI genes.** (A) Distribution of number GWAS hits in the 343 Individual GWAS studies we detected PPI genes in. (B) Distribution of PPI genes in the 343 individual GWAS studies we detected PPI genes in. (C) Scatter plot of number of GWAS hits in each study compared to the number of PPI genes in each study, showing very little correlation. (D) Distribution of the number of GWAS hits in 12 parent terms that we detect PPI genes in. (E) Distribution of PPI genes in the 12 parent terms that we detect PPI genes in. (F) Scatter plot of the number of GWAS hits in each parent term compared to the number of GWAS hits.
(TIF)

**S4 Fig. Example PPI genes.** PPI genes in Morning vs. Evening chronotype, Parkinson's Disease, A1C Measurement and Alzheimer's Disease. The GWAS catalog study accessions are shown within parentheses.
(TIF)

**S5 Fig. Analysis of Clinvar pathogenic variants stratified by degree bin.** Individual GWAS studies: after stratifying by degree-bin we found that PPI-GWAS had the highest proportion of Clinvar pathogenic variants across 4 degree bins (statistically significant in 2 bins) consistent with what was observed prior to stratifying by degree. Parent Term analysis: we found that PPI only had a greater proportion than GWAS only in 3 bins, and PPI-GWAS has a greater proportion than GWAS only in 2 bins.
(TIF)

**S6 Fig. Cancer parent term PPI genes.** PPI only are shown in yellow while PPI-GWAS are shown in green, and PPI edges are represented as lines. We used the STRING network to extract the directly interacting partners of each of the 109 cancer parent term PPI genes. Then

we did a pairwise comparison to count the number of directly interacting partners that are shared across each node pair, and then we used the number of shared interacting partners to weight the edge between the 2 nodes. We then used the Fruchterman Reingold force directed layout algorithm [60] within Gephi [59] to visualize the network.
(TIF)

**S7 Fig. Stratifying cancer parent term PPI genes by degree bin.** Similar to the unstratified analysis we found that PPI-GWAS and PPI only were enriched for COSMIC germline variants, and PPI-GWAS was enriched compared to GWAS only in Oncokb drug targets and PPI-G-WAS was enriched compared to GWAS only in Therapeutic Targets Database cancer drug targets.
(TIF)

**S8 Fig. Excess PPI among somatically mutated cancer genes.** The size of the node corresponds to the degree, and PPI are represented with lines.
(TIF)

**S9 Fig. Degree distribution of the PPI genes compared to the rest of the network.**
(TIFF)

**S10 Fig. Proportion of somatically mutated PPI genes in bins of different degree size.**
(TIF)

**S11 Fig. Somatically mutated cancer PPI genes stratified by degree bin.** Similar to the unstratified analysis we found that compared to Somatically Mutated, PPI-Somatically mutated was enriched for COSMIC somatic variants, and PPI-Somatically Mutated was enriched compared to Somatically Mutated in Oncokb drug targets.
(TIF)

**S1 Table. The PPI network used in our study.** There are 11,049 unique nodes in this network.
(XLSX)

**S2 Table. List of GWAS hits in each study.** This table shows the study accession, disease trait, parent term, GWAS hit, SNP, Ancestry, whether the study has excess PPI among GWAS hits and whether a candidate gene was detected, from the 1,381 GWAS studies that went into the analysis.
(XLSX)

**S3 Table. List of candidate core genes in the 343 GWAS studies.** This table contains the study accession, the disease trait, the candidate core gene, the GWAS hits that detected the candidate core gene, the raw p-value, adjusted p-value, Ancestry and whether the study had excess PPI edges between GWAS hits, and whether the candidate core genes is PPI only or PPI-GWAS.
(XLSX)

**S4 Table. List of genes within our network that contain Clinvar Pathogenic variants.**
(XLSX)

**S5 Table. Rank fractions calculated from GWAS p-values.** Rank fractions calculated from GWAS p-values for GWAS hits and PPI-GWAS from 119 GWAS studies that contain at least 1 PPI-GWAS. This table contains the rank, study accession, GWAS p-value, disease trait, GWAS hit, SNP, whether the GWAS hits is PPI-GWAS or not, the number of GWAS hits in

each study and the rank fraction.
(XLSX)

**S6 Table. List of candidate core genes in Parent Terms.** This table contains the Parent Term, candidate core gene, the GWAS hits that detect each candidate core gene, the raw p-value, adjusted p-value, and whether the candidate core gene is PPI only or PPI-GWAS.
(XLSX)

**S7 Table. PPI-GWAS vs. PPI only proportions across Parent Terms.** This table shows the proportion of PPI-GWAS vs. PPI only candidate core genes in each of the 12 parent terms. It shows that Lipid or Lipoprotein measurement has the highest proportion of PPI-GWAS.
(XLSX)

**S8 Table. Enriched pathways across Parent Terms.** Representative examples of disease relevant pathways enriched amongst parent term candidate core genes that contain Clinvar pathogenic variants.
(XLSX)

**S9 Table. List of the different Cancer studies in the Cancer parent term.** This table shows the number and type of the different cancer studies represented in all 142 GWAS studies that correspond to the Cancer parent term. It contains the number of GWAS hits from the cancer study as well as the number of studies from that cancer type. It demonstrates that Prostate Cancer and Breast Cancer are highly represented.
(XLSX)

**S10 Table. Enriched pathways in Cancer Parent Term.** Pathways enriched in the 78 PPI only Cancer Parent Term Candidate Core Genes, compared to the whole network (n = 11,049).
(XLSX)

**S11 Table. List of 95 COSMIC Germline genes that are also within our network.**
(XLSX)

**S12 Table. List of 52 Oncokb Cancer drug targets that are also within our network.**
(XLSX)

**S13 Table. List of 207 Cancer Drug Targets from the Therapeutic Targets Database, that are also present within our network.**
(XLSX)

**S14 Table. List of somatically mutated genes.** This table shows the 218 somatically mutated cancer genes identified in [55].
(XLSX)

**S15 Table. List of candidate core genes detected from somatically mutated cancer genes.** This table shows the candidate core genes that were identified from the 218 somatically mutated genes listed in S14 Table. It contains the Somatically Mutated candidate core gene, the number of PPI edges to the 218 somatically mutated genes, the raw p-value, the adjusted p-value, whether it is PPI only or PPI- Somatically Mutated and the list of Somatically Mutated genes that it overlaps with.
(XLSX)

**S16 Table. List of 472 COSMIC Somatic genes that are also within our network.**
(XLSX)

**S17 Table. Candidate core genes predicted across different types of networks.** Candidate core genes predicted using the 403 Cancer parent term GWAS hits across networks built from different interaction types (Reaction, Catalysis, Activation, Inhibition, Ptmod (post-translational modification), expression and coexpression).
(XLSX)

## Acknowledgments

We would like to acknowledge Ronglai Shen for helpful discussions regarding the design of some of the statistical tests used in the manuscript, and would like to thank Samuel Zimmerman for helpful feedback on an earlier version of the manuscript.

## Author Contributions

**Conceptualization:** Abhirami Ratnakumar.

**Data curation:** Abhirami Ratnakumar.

**Formal analysis:** Abhirami Ratnakumar, Jessica C. Mar, Nadeem Riaz.

**Funding acquisition:** Jessica C. Mar, Nadeem Riaz.

**Investigation:** Abhirami Ratnakumar, Jessica C. Mar, Nadeem Riaz.

**Methodology:** Abhirami Ratnakumar, Jessica C. Mar, Nadeem Riaz.

**Supervision:** Jessica C. Mar, Nadeem Riaz.

**Visualization:** Abhirami Ratnakumar, Nils Weinhold, Jessica C. Mar, Nadeem Riaz.

**Writing – original draft:** Abhirami Ratnakumar.

**Writing – review & editing:** Abhirami Ratnakumar, Nils Weinhold, Jessica C. Mar, Nadeem Riaz.

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
