## [Decision Letter · Decision Letter 0]

3 Nov 2019

Dear Dr Ratnakumar,

Thank you very much for submitting your Research Article entitled 'Protein-Protein interactions uncover ‘core genes’ within Omnigenic disease networks' to PLOS Genetics. Your manuscript was fully evaluated at the editorial level and by independent peer reviewers. The reviewers appreciated the attention to an important problem, but raised some substantial concerns about the current manuscript. Based on the reviews, we will not be able to accept this version of the manuscript, but we would be willing to review again a much-revised version. We cannot, of course, promise publication at that time.

If you decide to revise the manuscript for further consideration at PLOS Genetics, please aim to resubmit within the next 60 days, unless it will take extra time to address the concerns of the reviewers, in which case we would appreciate an expected resubmission date by email to plosgenetics@plos.org.

[LINK]

We are sorry that we cannot be more positive about your manuscript at this stage. Please do not hesitate to contact us if you have any concerns or questions.

Yours sincerely,

Xuanyao Liu, Ph.D.

Guest Editor

PLOS Genetics

Gregory Barsh

Editor-in-Chief

PLOS Genetics

Although the reviewers commented favorably on the general goal and approach of the manuscript, there were significant criticisms that preclude publications. These comments indicate that significant amount of re-analysis is need to make the paper convincing (See comments of reviewer1 and reviewer2). "Core genes" or the gene targets that your approach aims to identify need to be carefully defined, and additional analyses need to support your claims on the findings. We would be willing to reconsider the manuscript after it has undergone a major revision that takes into account of the criticisms of the reviewers, with no assurance of acceptance.

Reviewer's Responses to Questions

**Comments to the Authors:**

Reviewer #1: In this paper, the authors present a method to identify genes which are enriched for protein-protein interactions (ppi) with genes near GWAS loci (or some other list of genes). The authors argue that these represent “core genes” under the omnigenic model. They describe some examples where the identified genes are known to be highly disease-relevant, and they report that their list of genes is enriched for signal according to clinvar and according to pLI.

I tend to think the authors are onto something with this method, but more work is needed before the paper is convincing. In particular, the authors must distinguish between the following claims: (1) genes with high ppi to GWAS-proximal genes, but without GWAS hits themselves (ppi-only genes), are bonafide disease-relevant genes. (2) among GWAS-proximal genes, those with ppi to other GWAS-proximal genes (ppi-gwas genes) are more disease relevant than those without (gwas-only genes). (3) ppi-only genes, in addition to being disease relevant, are even more disease relevant than gwas-only genes.

This is the sequence of claims that is necessary to justify calling the ppi genes “core”, implying that they are more important than most GWAS genes, which are presumably peripheral. The authors have provided evidence for (1), but not for (2-3). I have some suggestions for how they could make these comparisons. However, I do not think that acceptance of the paper should be contingent on identifying strong evidence for both these claims (see below).

The ClinVar and pLI comparisons are made between ppi genes (including both ppi-gwas and ppi-only) and all genes. There is also a comparison with “parent” genes, i.e. genes with high ppi to the larger set of parent-term GWAS-proximal hits. Missing is a comparison with gwas-only genes. I would like to see a comparison between ppi-gwas and gwas-only, and between ppi -only and gwas-only. These comparisons could provide some evidence of (2) and (3) respectively. Moreover, are ppi-only genes enriched for heritability, even if thehy do not harbor significant GWAS hits? Are ppi-gwas genes more likely to be among the very most significant genes, compared with gwas-only genes? For each parent category, is the ClinVar enrichment signal concentrated among genes associated with parent-category-related pathologies?

Even if all these analyses are positive, a critical caveat is that many GWAS-proximal genes will not actually be the right gene. (Could it be possible to observe this empirically, by showing that the 2nd closest gene to each GWAS hit is also enriched with ppi to other GWAS-proximal genes and this cannot be explained by proximal genes having shared ppi?) The reason this caveat can affect these analyses is that it is difficult to distinguish between two possibilities: (1) ppi-gwas genes are more likely to have ClinVar phenotypes because they are more important for disease than gwas-only genes, even though the gwas-only genes are also bona fide disease genes; and (2) ppi-gwas genes are more likely to have ClinVar phenotypes because they are more likely to be the true disease gene, whereas gwas-only genes are more likely to be the wrong gene at the locus altogether (but the correct gene, if we knew it, would be just as important as a ppi-gwas gene). I think the authors need to discuss this distinction. I don’t see a way that they can realistically rule out possibility 2, but that’s ok: their list of genes is interesting under either possibility 1 or possibility 2.

If the suggested analyses are negative, I don’t think it should necessarily preclude publication of this paper, but I would think that some re-writing would be needed. At a minimum, “core genes” should be replaced with “GWAS-interacting gene” or some other, less aspirational term. It’s still fine to motivate the network analyses using the omnigenic model, and say that your goal is to identify core genes, but if there is evidence that the identified genes are any more “core” than other GWAS genes, then it doesn’t seem appropriate to call them “core genes”. You can call p53 a core gene in cancer, but it doesn’t follow that all the other ppi genes are just as important.

I want to emphasize again that it is still valuable to identify new disease-relevant genes via ppi with other disease genes, even if you aren’t claiming that they are all “core” or even enriched for core genes relative to other GWAS genes.

I have some minor comments as well:

1. I noticed that many HLA genes are detected, and usually they are annotated as “new” even for phenotypes with clear HLA associations. I think there may be some issue at HLA where only one gene is being picked as the GWAS gene, and the others are being called “new” even though this obviously isn’t appropriate at this locus. Probably change the “new” criterion to “not within x Kb of the lead SNP”, and possibly exclude HLA from all analyses.

2. The omnigenic model is usually described in terms of gene-regulatory networks, not ppi networks. I don’t think this distinction is critical to the model, but maybe the authors could clarify that ppi is only one way that genes can interact, that it is not the type of interaction most people would associate with the omnigenic model, but that it is just as plausible for peripheral gene effects to be mediated by ppi as it is for them to be mediated by gene-regulatory networks. In addition, the authors could either apply their method to some gene-regulatory networks or explain why they have chosen not to do so (I don’t actually recommend doing this).

3. For any genes that you especially highlight in the text (BRCA1, APP, etc), make sure to be clear about whether they are annotated as GWAS genes in your dataset and also about whether they have been implicated by genetic studies that you did not include. I think a main table could be helpful here: for example, the BRCA1 row could contain a p-value using your method, the rank of this p-value out of all genes, a “no” indicating that this gene is not annotated as a cancer gene in your GWAS data set, and a reference for a study about rare variation in BRCA1.

4. The writing style is idiosyncratic, especially in the methods section. I found it perfectly readable, but it could be good to run this by someone who is a strong writer.

Reviewer #2: See attachment.

Reviewer #3: The manuscript by Ratnakumar and colleagues presented results analyzing GWAS associated genes in light of protein-protein interaction (PPI) networks. Their hypothesis is that genes physically interacting with disease-associated genes with high frequencies are candidates of "core genes", according to the omni-genetic model of complex trait genetics. Their analysis identified hundreds of putative "core genes" across a large set of phenotypes.

While I think some of the results, especially those about cancer, are interesting, I am not convinced that their analysis really sheds light on core genes under the omnigenetic model. Also the overall novelty of the work is questionable to me. Below are specific comments:

Major comments

1) Core genes have a specific meaning under the omnigenetic model: they are believed to mediate the effects of a potentially large number of genes associated with a complex trait (called "peripheral genes"). The omnigenetic paper discusses regulatory influence of perturbation of peripheral genes on core genes, in partiular, as one main mechanism. It seems to me that this paper mis-interprets the concept of core genes. All the analysis are based on the numbers of PPIs: there is no evidence that the genes interacting with other GWAS genes would "mediate" their effects on disease risks.

The analysis is more like finding "hub" genes in PPI network. There is a large literature on this topic, including some in the context of disease genetics, e.g. 21764832, 21490723, 25915600. It seems to me that none of these papers is properly cited. Given this background, I think the conceptual novelty of the work is quite limited.

2) In Figure 3d: the authors show that their core genes are more enriched with constrained genes (according to pLI scores). The authors compare pLI distribution of the core genes vs. all genes in the PPI network. I think the core genes are likley to have higher degress than background, so this analysis should control for degree distribution of two gene sets being compared. Otherwise, the results would show "highly connected" genes are more likely to have high pLI scores. This would be similar to earlier studies showing hub genes are under stronger evolutionary constraint, but this would have no implications on genetics of diseases.

3) The results of cancer core gene analysis are interesting. However, the analysis seems somewhat superficial. I would recommend the authors to perform some deeper analysis. For example, they reported >100 new genes, what are the overepresented biological pathways other than immune genes? Any specific new genes that may provide some new insights on cancer? If one compares across cancer types, are the genes usually found in one cancer type or multiple cancer types?

Minor comments

1) Figure legends are generally too long and sometimes redundant with the main text.

2) Lines 336-340: the authors mentioned that lipid traits have the highest proportion of core genes that are also top GWAS hits, and offered the possible explanation that lipid traits may be less polygenic. While this may well be true, I think it is important to consider the power of studies. For studies with large sample sizes (likely the case for lipid traits), it is likely that many "core" genes would be found as GWAS hits, so the proportion would be high.

3) Line 358: should cite Figure 3e instead of 3c.

**Have all data underlying the figures and results presented in the manuscript been provided?**

Reviewer #1: Yes

Reviewer #2: Yes

Reviewer #3: Yes

PLOS authors have the option to publish the peer review history of their article (what does this mean?). If published, this will include your full peer review and any attached files.

Reviewer #1: Yes: Luke O'Connor

Reviewer #2: No

Reviewer #3: No

---

## [Decision Letter · Decision Letter 1]

30 Mar 2020

Dear Dr Ratnakumar,

Thank you very much for submitting your Research Article entitled 'Protein-Protein interactions uncover candidate ‘core genes’ within Omnigenic disease networks' to PLOS Genetics. Your manuscript was fully evaluated at the editorial level and by independent peer reviewers. While we cannot make a definite commitment, we will probably accept your paper for publication, if you provide an answer to reviewer 1 on the node degree correction for Clinvar analysis and the drug target analysis, and make changes according to the remaining of the reviewers’ comments.

[LINK]

Yours sincerely,

Xuanyao Liu, Ph.D.

Guest Editor

PLOS Genetics

Gregory Barsh

Editor-in-Chief

PLOS Genetics

Reviewer's Responses to Questions

**Comments to the Authors:**

Reviewer #1: The authors have provided a thorough response to my comments and suggestions. Following the comments of Reviewer 2, I am now curious whether the Clinvar analysis and the drug target analysis were corrected for node degree. Otherwise, I only have presentation-related comments:

• In the current presentation, it feels like you are assuming your hypothesis is true (that ppi genes are candidate core genes) throughout pages 9-15. I would recommend that in this section, you use the term “ppi genes” instead of “candidate core genes”. Then, you could retitle the section starting on page 15 “Clinvar enrichment: ppi genes are candidate core genes.” The title for p. 21 could be “Drug target enrichment: ppi genes are candidate core genes”, and on p. 22, “Cancer driver enrichment: …”.

• The main figures should be readable without referencing the caption. Currently, many axes are unlabeled, and the titles are inconsistent (Figures 3-5). I would recommend adding axis labels everywhere it is not totally obvious and making all titles describe what is being plotted (rather than delivering the punchline).

• Consider including the finding of no excess GWAS signal at ppi-only genes. This analysis suggests that ppi-only genes would not be detected in GWAS of larger sample size.

• In many places, a long sentence should be split into two sentences. For example, see lines 85-89, 104-109, 202-205, 250-259, 637-640. In a number of places, a comma should be deleted (e.g. lines 637, 644, 139).

Reviewer #2: I congratulate the authors for successfully incorporating reviewer feedback into the manuscript. My concerns with the text have been satisfied, and I have only minor comments below.

Line 65: “insulin for” rather than “insulin in”

Line 66: BRCA1, as a protein, should not be italicized

Line 69: Technically the experiments still have to be performed. It is more accurate to say “by utilizing publicly available reference datasets.”

Line 264: delete the second comma

Line 386-389: Sentence is difficult to make sense of.

Line 473: delete the comma

Line 573-586: I think this could be condensed to provide a simple, short summary.

Line 597-603: I think it suffices to say “We present a new application of GWAS data: identifying core genes as envisioned by the omnigenic model that do not themselves exhibit GWAS signal by consulting physical interactions.”

Line 629-631: Sentence should be rewritten.

Line 637: delete first comma.

Line 644-653: I recommend deleting this paragraph. It is not terribly helpful.

Line 666-667: As described earlier, the method does not exactly work on every GWAS – the GWAS must have enough hits with PPI edges. It would be better to say that this method adds another layer to interpreting GWAS signal.

Lines 897-906: This should be rewritten so say very clearly that candidate core genes within 1 Mb of a GWAS hit are PPI-GWAS hits while the rest are PPI only. Any GWAS hits without a candidate core gene are GWAS-only.

Lines 949-952: Delete.

Lines 1046-1050: Delete.

In general: The omnigenic model is by convention not capitalized

Figure 1c: gene symbol names and axis text too small

Figure 1b step 2: axis text way too small

Figure 3d,e: axis text too small

Figure 4e: axis text too small

Figure 5e: number text too small

Figure 5f: axis text too small

Reviewer #3: The reviewers have done a good job at addressing my comments. While conceptually related to several papers published in the past, I think the large-scale analysis the authors performed across many GWAS datasets would be a useful resource for the community.

**Have all data underlying the figures and results presented in the manuscript been provided?**

Reviewer #1: Yes

Reviewer #2: Yes

Reviewer #3: Yes

---

## [Decision Letter · Decision Letter 2]

1 Jun 2020

Dear Dr Ratnakumar,

We are pleased to inform you that your manuscript entitled "Protein-Protein interactions uncover candidate ‘core genes’ within Omnigenic disease networks" has been editorially accepted for publication in PLOS Genetics. Congratulations!

Yours sincerely,

Xuanyao Liu, Ph.D.

Guest Editor

PLOS Genetics

Gregory Barsh

Editor-in-Chief

PLOS Genetics

Comments from the reviewers (if applicable):

The authors have addressed all the comments of the reviewers to their satisfaction. Thank you very much for the effort.

Reviewer's Responses to Questions

**Comments to the Authors:**

Reviewer #1: The authors have fully addressed my comments.

Reviewer #2: I commend the authors for their diligence in responding to reviewer comments. My concerns have been fully addressed.

**Have all data underlying the figures and results presented in the manuscript been provided?**

Reviewer #1: Yes

Reviewer #2: Yes

PLOS authors have the option to publish the peer review history of their article (what does this mean?). If published, this will include your full peer review and any attached files.

Reviewer #1: Yes: Luke O'Connor

Reviewer #2: Yes: Evan A Boyle

**Data Deposition**

http://datadryad.org/submit?journalID=pgenetics&manu=PGENETICS-D-19-01600R2

**Press Queries**

---

## [Editor Report · Acceptance letter]

6 Jul 2020

PGENETICS-D-19-01600R2 

Protein-Protein interactions uncover candidate ‘core genes’ within omnigenic disease networks 

Dear Dr Ratnakumar, 

We are pleased to inform you that your manuscript entitled "Protein-Protein interactions uncover candidate ‘core genes’ within omnigenic disease networks" has been formally accepted for publication in PLOS Genetics! Your manuscript is now with our production department and you will be notified of the publication date in due course.

With kind regards,

Matt Lyles

PLOS Genetics

On behalf of:
